# Re-order parameter of interacting thermodynamic magnets

Byung Cheol Park[1,2,5], Howon Lee[3,4,5], Sang Hyup Oh[3], Hyun Jun Shin ®[3], Young Jai Choi ®[3] ✉ & Taewoo Ha ®[1,2] ✉

Phase diagrams of materials are typically based on a static order parameter, but it faces challenges when distinguishing subtle phase changes, such as re-ordering. Here, we report a dynamic nonequilibrium order parameter termed re-order parameter to determine subtle phases and their transitions in interacting magnets. The dynamical precession of magnetization, so-called magnon, premises as a reliable re-order parameter of strong spin-orbit coupled magnets. We employ orthoferrites $YFeO_3$ and its Mn-doped variations, where diverse magnetic phases, including canted antiferromagnetic ($\Gamma_4$) and collinear antiferromagnetic ($\Gamma_1$) states, have been well-established. Low-energy magnon uncovers the spin-orbit coupling-induced subtle magnetic structures, resulting in distinct terahertz emissions. The temporal and spectral parameters of magnon emission exhibit characteristics akin to BCS-type order parameters, constructing the magnetic phase diagram of Mn-doped $YFeO_3$. This approach further reveals a concealed ferrimagnetic phase within the $\Gamma_1$ state, underscoring its potential to search for hidden phases of materials, completing their phase diagrams.

A phase diagram represents diverse states of a substance, based on external factors such as temperature and pressure[1,2]. This is vital for decision-making, materials design, and matter understanding. An order parameter, a measure of the degree of order across the boundaries in a phase transition system[1,2], serves as a comprehensive determinant of matter phases in classical thermodynamics, such as magnetization in ferromagnets and polarization in ferroelectrics[3,4]. They effectively distinguish the ordered state, characterized by a maximum saturated value, from the disordered state, where the value is zero[3,4]. Complex interactions within ordered states give rise to multiple distinct phases. Then, classical order parameters face challenges in distinguishing these phases with subtle changes, such as re-ordering. This necessitates the development of novel parameters to classify the nuanced phases emerging from interactions like electron correlation and spin-orbit coupling.

Magnetic materials primarily divided into two categories: ferromagnets (FMs), where aligned spins create net magnetization (**M**), and antiferromagnets (AFMs), where antiparallel spin alignment yields zero **M** (refs. 3,4). This **M** has served as the equilibrium order parameter that distinguishes these core magnetic phases. Temperature-dependent **M**($T$) unveils ferromagnetic transitions with rotational symmetry breaking—**M** sharply rises from null at the transition temperature $T_c$, showing the first-order transition (Fig. 1a) (refs. 3,4). However, due to spin-orbit coupling (SOC) that causes spin canting, AFMs also can exhibit net **M** (refs. 5,6), introducing intricacy to the phase diagram and casting doubt on the classical order parameter. Indeed, the conventional equilibrium parameter **M**, acquired through static intensity measurements, such as magneto-metry faces challenges in categorizing subtle magnetic structures, e.g., between canted AFM and FM.

---

[1]Sungkyunkwan University, Suwon 16419, Republic of Korea. [2]Center for Integrated Nanostructure Physics, Institute for Basic Science, Sungkyunkwan University, Suwon 16419, Republic of Korea. [3]Department of Physics, Yonsei University, Seoul 03722, Republic of Korea. [4]Present address: Center for Spintronics, Korea Institute of Science and Technology (KIST), Seoul 02792, Republic of Korea. [5]These authors contributed equally: Byung Cheol Park, Howon Lee. ✉e-mail: phylove@yonsei.ac.kr; bspha77@gmail.com

**a**

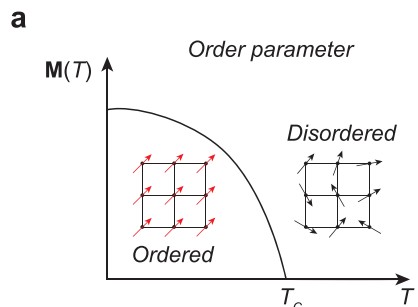

**b**

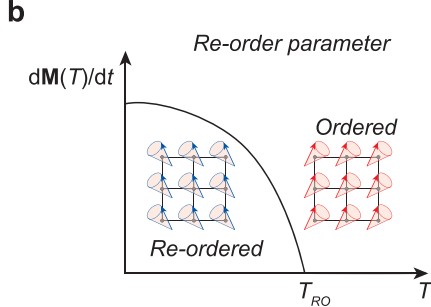

**Fig. 1 | Introduction of re-order parameter. a** Classical order parameter in thermal equilibrium. As temperature ($T$) decreases, the disordered state transitions into an ordered state through the alignment of the thermodynamic order parameter, namely magnetization $\mathbf{M}(T)$ for ferromagnets. The order parameter exhibits a first-order-like transition at the critical temperature $T_c$, which can be effectively described by a BCS-type transition curve. **b** Dynamic re-order parameter in nonequilibrium. As $T$ decreases, the ordered state undergoes re-ordering at the re-ordering temperature $T_{RO}$, accompanied by a first-order-like transition in the re-order parameter, denoted as $d\mathbf{M}(T)/dt$. This re-order parameter, responsible for the precession of magnetization in magnets, demonstrates a high sensitivity to subtle re-ordered phases.

This necessitates the development of a new order parameter extending beyond static equilibrium, offering a credible alternative not only suitable for classical magnets but also capable of accommodating interaction effects. SOC induces subtle changes in spin ordering, such as the emergence of canted AFM phases with net $\mathbf{M}$ (refs. [5,6]), resulting in distinct magnetization dynamics. Interestingly, the observation of the magnon within various magnetic phases suggests their potential utility in distinguishing these subtle magnetic states[7–11]. Despite their clear association with both static spin arrangement and spin canting, magnons have yet to be recognized as the order parameter.

We propose that magnons, representing the precession of $\mathbf{M}(T)$, assume the role of a re-order parameter (Fig. 1b), effectively delineating first-order-like magnetic transitions from the ordered state to the re-ordered state. An exceptional feature of this re-order parameter lies in its ability to directly excite magnons, excluding other spin effects stemming from inevitable sample imperfections, such as magnetic boundaries and clusters, aiding magnetic phase diagram studies, even in polycrystalline magnets. The counterintuitive notion that magnons, as dynamical modulation of the conventional order parameter $\mathbf{M}$, determine magnetic phases, is remarkably unexpected.

## Results

Orthoferrite YFeO$_3$, characterized by an orthorhombic perovskite structure (space group Pbnm)[12–15], can serve as a testbed for validating magnons as the re-order parameter. Strong SOC in these materials guarantees rich magnetic phases across the canted AFM ($\Gamma_4$) and collinear AFM ($\Gamma_1$) phases[12–15]. Figure 2a depicts a canted AFM spin ordering of YFeO$_3$ below Néel temperature $T_N = 645$ K (refs. [12–15]). The complex spin ordering can be simplified with two magnetizations $\mathbf{M}_1$ and $\mathbf{M}_2$. The $\Gamma_4$ phase manifests a weak FM moment from canting of $\mathbf{M}_1$ and $\mathbf{M}_2$ along the c-axis due to SOC, which persists below $T_N$ without applying an external magnetic field along the a-axis that incudes spin-reorientation transition[16].

The presence of spin canting in orthoferrites inevitably results in the coexistence of quasi-FM and quasi-AFM magnons, as illustrated in Fig. 2b. The quasi-FM mode arises from the precession of $\mathbf{M}_1$ and $\mathbf{M}_2$ with a phase difference of $\pi/2$, while the quasi-AFM mode is attributed to precession without any phase difference. The introduction of transition metal ions into the Fe$^{3+}$ sites is known to induce spin-reoriented phases, exemplified by the substitution of Mn$^{3+}$ causing the Morin-type spin-reorientation transition from the $\Gamma_4$ phase to the collinear AFM phase $\Gamma_1$ (refs. [17–20]). Naturally, this phase transition is supposed to change the dynamics of magnons.

By circumventing other charge effects, terahertz (THz) excitation facilitates low-energy magnon excitation in semiconducting orthoferrites. Notably, their substantial band gap and lack of free carriers or low-energy phonons underscore THz field generation of magnon that serves as the re-order parameter. As described in Fig. 2c, the magnetic field component ($\mathbf{H}_{THz}$) of the THz transient applies torque to the net moment, inducing precession and resulting in THz electromagnetic field emission during relaxation, governed by the Landau–Lifshitz–Gilbert equation[21]:

$$\frac{d\mathbf{M}}{dt} = -|\gamma|(\mathbf{M} \times \mathbf{H}_{eff}) + \frac{\alpha}{\mathbf{M}}\left(\mathbf{M} \times \frac{d\mathbf{M}}{dt}\right) \quad (1)$$

with gyromagnetic ratio $|\gamma|$ and Gilbert damping $\alpha$. Featuring SOC-induced spin canting, orthoferrites possess both THz quasi-FM and quasi-AFM magnon modes, distinct from collinear AFMs.

Time-resolved THz spectroscopy accurately captures magnon dynamics in picoseconds. Figure 2d (top panel) presents THz electric ($\mathbf{E}_{THz}$) fields—a primary pulse remains after magnon generation and is followed by magnon emission. Although the $\mathbf{H}_{THz}$-field component triggers magnon excitation, the resulting emission registers as an $\mathbf{E}_{THz}$-field. In the case of the parent compound YFeO$_3$, two distinctive magnons associated with the canted AFM are observed in the time-domain as a superposition of two oscillatory signals with distinct frequencies (inset of Fig. 2d (top panel)).

Next, the transmission spectrum is obtained through the Fourier transformation of the complete time-domain signal (comprising the primary pulsed $\mathbf{E}_{THz}$-field and the oscillatory magnon emission), whereas the emission spectrum exclusively utilizes the oscillatory magnon emission that follows the primary pulsed $\mathbf{E}_{THz}$-field. These spectra consistently reveal a quasi-FM magnon at 10 cm$^{-1}$ ($\approx 1.24$ meV $\approx 0.3$ THz) and a quasi-AFM magnon at 18 cm$^{-1}$ ($\approx 2.23$ meV $\approx 0.54$ THz) as consistent with the previous works[7,8,10,20]. We emphasize that the nature of the sample, whether it is a single crystal or polycrystal, does not affect magnon observation.

The spectral analysis (Fig. 2e) unveils key magnon parameters, including the precession frequency ($\omega_0$), magnon density ($\omega_p^2$), and scattering rate ($\gamma$) (or equivalently, inverse lifetime $\gamma = 1/\tau$). These parameters are acquired from the Lorentzian model fitting to the quasi-FM magnon conductivity spectrum (left inset of Fig. 2e) (see Supplementary Figs. 1–3 for fitting details). For low-doped YFe$_{0.97}$Mn$_{0.03}$O$_3$, $\omega_p^2$ presents no clear change over the temperatures (4 K $\leq T \leq$ 295 K). In contrast, such temperature dependence of $\omega_p^2$ distinctly depicts a first-order-like transition for high-doped YFe$_{0.85}$Mn$_{0.15}$O$_3$, which is reminiscent of order parameter $\mathbf{M}$ described by BCS-type transition (Methods). The magnetic transition consistently adheres to the BCS model (Methods) as well as the power law in mean-field theory (order parameter $\propto (1 - T/T_c)^\beta$). However,

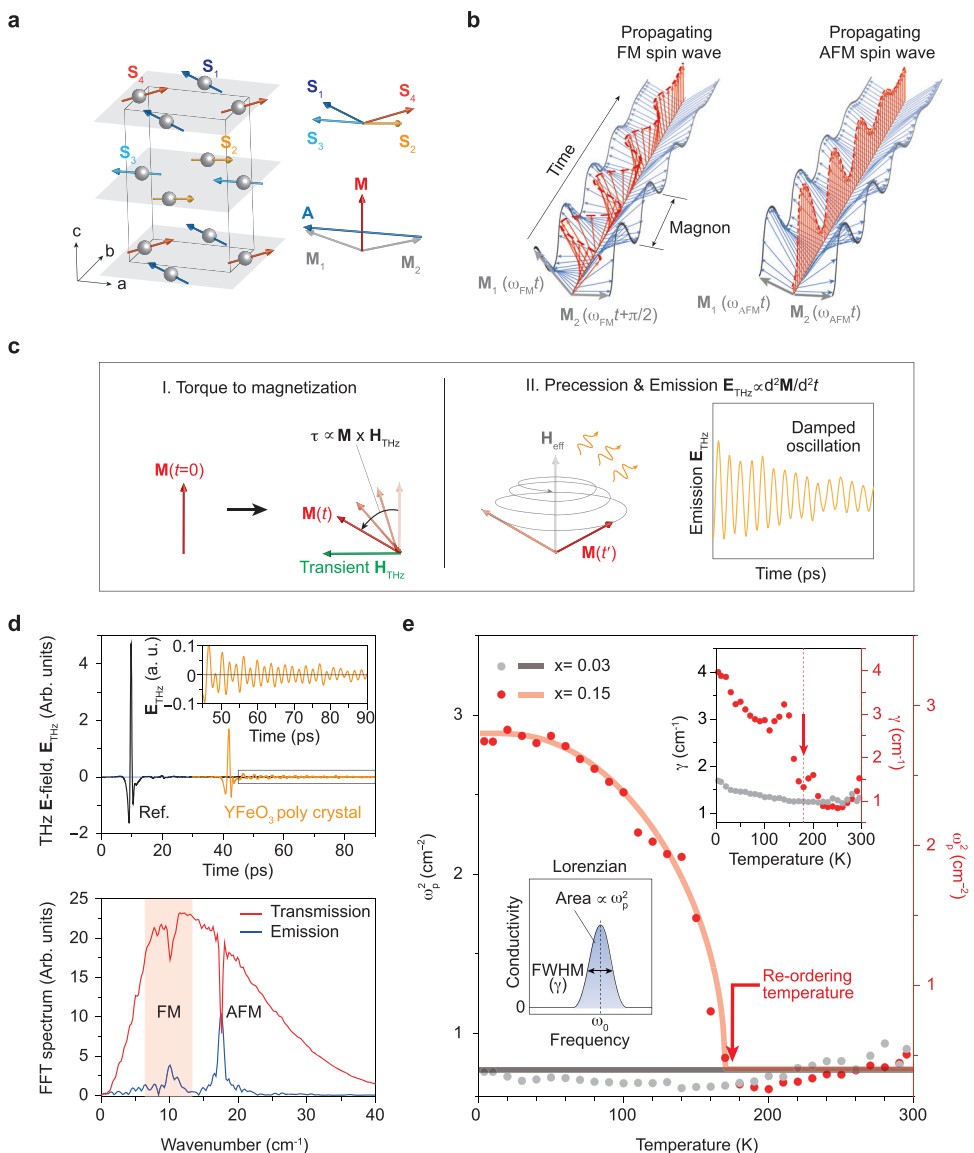

**Fig. 2 | Spin waves as the re-order parameter in canted antiferromagnetic (AFM) YFeO₃. a** Magnetic ordering of $YFeO_3$ below $T_N$. The magnetic ordering can be simplified by sublattice spins ($S_1$, $S_2$, $S_3$, $S_4$) and further by magnetization $M_1(= S_1 + S_3)$ and $M_2(= S_2 + S_4)$, resulting in the net $M(= M_1 + M_2)$ and Néel vector $A(= M_1 - M_2)$. **b** Two types of spin waves in $YFeO_3$. The quasi-FM spin wave (left) is a precession of net $M$, represented as a sum of $M_1$ and $M_2$ with a phase difference of π/2. The quasi-AFM spin wave (right) is a precession of net $M$ with in-phase $M_1$ and $M_2$. $\omega_{FM}$ and $\omega_{AFM}$ are the precession frequencies of quasi-FM and quasi-AFM spin waves, respectively. **c** Terahertz (THz) field-excitation of spin waves: (I) $M$ is rotated by a torque induced by the THz magnetic field ($H_{THz}$), resulting in the excitation of spin waves. (II) The excited spin wave radiates THz E-field ($E_{THz}$) during relaxation, captured by a THz detector. **d** Time-domain signal and its Fourier-transformed spectrum. In the time domain, the sample signal (orange) transmitted through the sample exhibits a delay relative to the reference signal (black) transmitted through

the blank holder. The sample signal (top panel) consists of the primary pulsed $E_{THz}$-field and the oscillatory $E_{THz}$-field from the magnon emission. The inset shows the magnified magnon emission signal. The transmission spectrum (red curve) of the entire time-domain signal and the emission spectrum (blue curve) of the oscillatory spin wave emission reveal a quasi-ferromagnetic spin wave at 10 cm⁻¹ ($\approx 1.24$ meV $\approx 0.3$ THz) and a quasi-AFM spin wave at 18 cm⁻¹ ($\approx 2.23$ meV $\approx 0.54$ THz). **e** Magnon density ($\omega_p^2$) serving as the re-order parameter. The $\omega_p^2$ of the quasi-FM magnon mode is acquired from the Lorentzian model fitting to the quasi-FM magnon conductivity spectrum (left inset). Dots represent experimental data, while solid lines represent fitting lines. No phase transition occurs for low-doped $YFe_{0.97}Mn_{0.03}O_3$ (black), while a first-order-like transition appears for high-doped $YFe_{0.85}Mn_{0.15}O_3$ (red), as described by a BCS-type transition curve (red line). The auxiliary re-order parameter, scattering rate ($\gamma$, inverse lifetime), shows a peak at the re-ordering temperature (right inset).

considering the significance of the saturation value reflecting the magnon density, we opted for the BCS model. This choice not only provides the transition temperature but also furnishes the saturation value, in contrast to the power law, which only identifies the transition temperature.

The transition temperature of $\omega_p^2$ closely coincides with the spin reorientation temperature of $YFe_{0.85}Mn_{0.15}O_3$ (ref. 18), highlighting the potential of $\omega_p^2$ as a reliable re-order parameter. Indeed, $E_{THz} \propto d^2M/dt^2$ for FM magnons[22] guarantees $dM/dt$ as an integration of $E_{THz}$

proportional to the strength of the magnon emission peak ($\propto \omega_p^2$). Note that the quasi-AFM magnon does not exhibit any order parameter-like features (See Supplementary Fig. 4). Moreover, $\gamma$ also exhibits a distinct peak at the re-ordering temperature (right inset of Fig. 2e), suggesting an auxiliary re-order parameter complementing the primary re-order parameter $\omega_p^2$.

We present a comprehensive phase diagram encompassing both the parent compound $YFeO_3$ and its Mn-doped variants ($YFe_{1-x}Mn_xO_3$), utilizing the re-order parameter $\omega_p^2$ (Fig. 3a). The auxiliary re-order

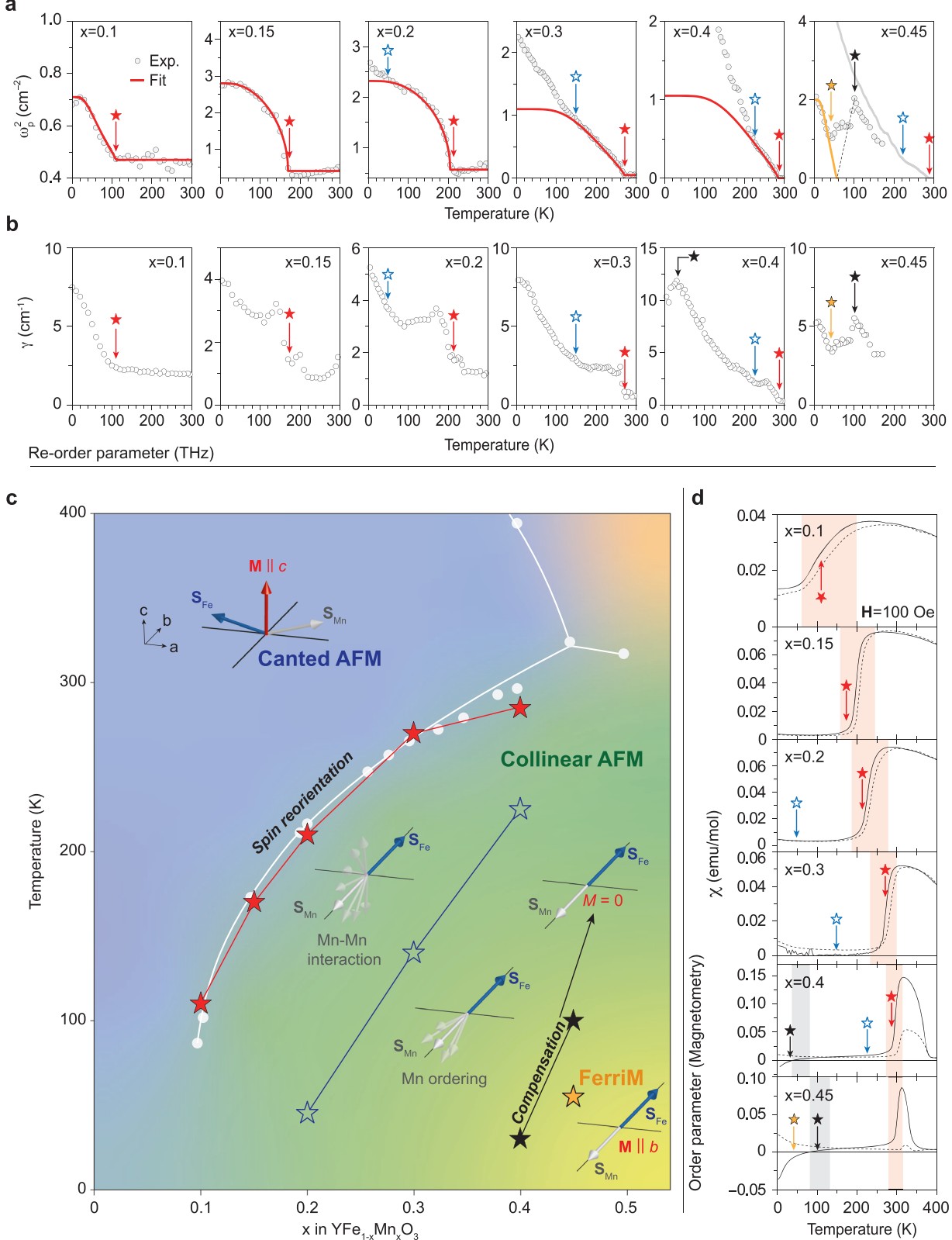

parameter $\gamma$ (Fig. 3b), as a complementary of re-order parameter, aids in comprehending the re-ordering process and the subtle variations in magnetization induced by Mn doping. Within the diagram, the phase boundaries are determined by tracking the transition temperatures of quasi-FM magnons, with respect to varying doping levels ($x$). The comprehensive phase diagram (Fig. 3c) reveals numerous intricate phases, encompassing the Mn fluctuation regime, Mn ordering regime,

magnetic compensation boundary, and ferrimagnetic (FerriM) phase. These findings significantly expand upon previous research, which primarily focused on the simplistic depiction of two phases: canted AFM and collinear AFM, surrounding the spin reorientation boundary (white circles in Fig. 3c, ref. 18).

First, the undoped and lightly doped samples (up to $x = 0.08$) exhibit no discernible phase transitions in the re-order parameter $\omega_p^2$

**Fig. 3 | Establishing a phase diagram using the re-order parameter for Mn-Doped YFeO₃.** **a** Temperature-dependent re-order parameter $\omega_p^2$ for Mn-doped YFeO₃ (YFe$_{1-x}$Mn$_x$O₃). Experimental data are represented by dots, while the BCS model fitting results are shown as lines. For $0.1 \leq x \leq 0.4$, red stars indicate the first-order-like transition temperatures, and blue stars correspond to temperatures where the experimental data begin to deviate from BCS traits. For $x = 0.45$, black and orange stars indicate indistinguishable points. Above 180 K, no experimental data are available due to significant broadening of magnon peak, and a grey line represents the experimental data for $x = 0.4$. **b** Temperature-dependent auxiliary re-order parameter $\gamma$. The stars correspond to those in a. **c** Magnetic phase diagram, constructed using our re-order parameter data (stars). Results from magnetometry in reference (white circles) are also plotted for comparison. The phase diagram

encompasses the canted antiferromagnetic, collinear antiferromagnetic, and ferrimagnetic phases. The analysis of the re-order parameter reveals spin reorientation, magnetization fluctuation, and ordering due to Mn doping, as well as magnetization compensation between phases. The figure illustrates the spin arrangement of each phase, with a blue arrow indicating the Fe moment, a grey arrow representing the Mn moment, and a red arrow denoting the net moment. **d** Temperature-dependent magnetic susceptibility data for the Mn-doped YFeO₃ samples identical to those used in THz experiments. Shaded regions indicate the temperature of phase boundaries, determined by magnetic susceptibility. A constant external magnetic field (**H**) of 100 Oe is applied in this magnetometry experiment.

(See Supplementary Fig. 1). This suggests that undoped YFeO₃ and lightly doped YFe$_{1-x}$Mn$_x$O₃ exhibit no discernible alterations in magnetic ordering without spin reorientation, spin fluctuations, or magnetic compensation. In contrast, highly doped YFe$_{1-x}$Mn$_x$O₃ ($0.1 \leq x \leq 0.45$) reveal phase transitions in the re-order parameter $\omega_p^2$ (See Supplementary Fig. 2). Additional noteworthy observation is that the crossover temperatures tend to increase with higher values of $x$. For the range $0.1 \leq x \leq 0.4$, first-order-like transitions of re-order parameter $\omega_p^2$ are evident at re-ordering temperatures (red stars in Fig. 3), all of which are well-described by the BCS model fitting (Fig. 3a).

For highly doped YFe$_{1-x}$Mn$_x$O₃, these re-ordering temperatures closely coincide with the Morin-type spin reorientation temperatures (denoted by the white circles in Fig. 3c, ref. 18), signifying the transition of AFM ordering from the **a**-axis to the **b**-axis and the elimination of canting, as reported in previous studies[17–20]. This consistency serves to validate the robustness and reliability of the re-order parameter $\omega_p^2$. Moreover, the scattering rate $\gamma$ of magnons (Fig. 3b) experiences an increase of more than twofold at the re-ordering temperatures. This elevated $\gamma$ implies the disruption of magnon coherence, which could be associated with the fluctuation in the magnetization of Mn (**S**$_{Mn}$) on the background of the magnetization of Fe (**S**$_{Fe}$). Therefore, the $\gamma$ can serve as an auxiliary re-order parameter.

At lower temperatures (marked by blue stars in Fig. 3a), a significant deviation from the BCS characteristics induced by spin reorientation occurs. These previously unreported transitions raise intriguing questions. It is plausible that this deviation results from the influence of Mn doping, which alters the Mn-Mn interaction, ultimately leading to Mn ordering at lower temperatures. As Mn ordering takes place, the concurrent increase in both $\omega_p^2$ and $\gamma$ signifies an augmentation in magnon density but with a diminished level of magnon coherence. We also find an occurrence of crossing near Mn ordering temperatures (blue stars) in the two-dimensional plot of magnon emission with the $x$-axis (temperature) and $y$-axis (time), signaling the change in the relaxation time $\tau$ (See Supplementary Fig. 5).

In the case of $x = 0.45$, distinctive transitions (marked by black and orange stars in Fig. 3a,b) become apparent along with Mn ordering at lower temperatures. However, experimental data above 180 K are unobtainable due to the substantial peak broadening (i.e., the substantial magnon scattering rate $\gamma$). It is worth noting that the re-order parameter $\omega_p^2$ exhibits a sudden decrease below 100 K, signifying a reduction in magnon density due to magnetization compensation (black stars), where **S**$_{Fe}$ and **S**$_{Mn}$ offset each other. This compensation can be interpreted as an intermediate state in the Mn ordering process, transitioning towards a ferrimagnetic state (orange star) characterized by perfect collinearity between **S**$_{Fe}$ and **S**$_{Mn}$.

Indeed, our investigation unveils the presence of the FerriM phase deep within the collinear AFM phase for $x = 0.45$, as indicated by the subsequent rise in the re-order parameter $\omega_p^2$. As the magnetization (**M**) saturates, the magnon density ($\propto \omega_p^2$) experiences a sharp increase below the compensation temperature, aligning with expectations. Notably, the distinct characteristics of magnon emissions, as revealed by the (auxiliary) re-order parameters, offer precise identification of

the FerriM phase. This is particularly important since the FerriM phase may appear somewhat ambiguous in magnetometry results, which typically exhibit negative magnetic susceptibility[18,23].

To highlight the strength of (auxiliary) re-order parameters, we compare the phase identification based on the re-order parameter with that based on the conventional order parameter. Figure 3d presents temperature-dependent magnetic susceptibility data, where an external magnetic field (**H**) of 100 Oe is applied to the Mn-doped YFeO₃ samples used in the THz experiments.

Foremost, the alignment between THz results and magnetometry data reinforces the credibility of the re-order parameter. Around the spin reorientation temperature (marked by the red stars), both the magnetic susceptibility and THz data consistently exhibit a pronounced decrease with decreasing temperature. Additionally, a modest rise in magnetic susceptibility can be interpreted as indicative of Mn ordering (indicated by blue stars), whereas a null magnetic susceptibility points to magnetization compensation (black stars). While negative susceptibility alone may not serve as direct evidence for the FerriM phase (noted by the orange star), its combined observation alongside the sharp increase in the re-order parameter $\omega_p^2$ provides compelling support for the FerriM phase.

The pivotal distinction lies in the precision of the re-order parameter in discerning phases and identifying phase boundaries, as it provides sharp and well-defined transition temperatures. In comparison, magnetometry data displays a broad transition temperature range (shaded regions in Fig. 3d) with considerable error bars, often exceeding 50 K. Magnetic susceptibility involves extensive trial and error with varying magnetic fields to achieve accurate measurements, whereas re-order parameters do not require the application of magnetic fields. Correcting issues like the splitting of **M** during field cooling and heating cycles (Fig. 3d) and accounting for magnetic domain boundaries and clusters can be challenging. Relying solely on magnetic susceptibility makes it challenging to identify complex magnetic phases beyond the spin reorientation boundary, such as the Mn fluctuation regime, Mn ordering regime, compensation boundary, and FerriM phase.

## Discussion
We conclude this report by offering a perspective on dynamic order parameters, as depicted in Fig. 4. Dynamic order parameters, exemplified by d**M**/d$t$, serve a dual purpose, functioning as both classical order parameters and re-order parameters. Beyond the re-order parameter (main text), d**M**/d$t$ for FM magnons can also act as an order parameter: it remains at d**M**/d$t = 0$ in the disordered state (**M** $= 0$) and exhibits a sharp increase as **M** increases in the ordered state. The phases arising from ordering and re-ordering transitions can be understood through Landau free energy landscapes, where the minimum position dictates the dynamic order parameter value. With time-resolved measurements, time-domain oscillatory signals are responsible for the notable changes in d**M**/d$t$ across the ordering and re-ordering temperatures. The spectral area, proportional to the dynamic order parameter (e.g., $\omega_p^2 \propto$ d**M**/d$t$), can identify the nuanced

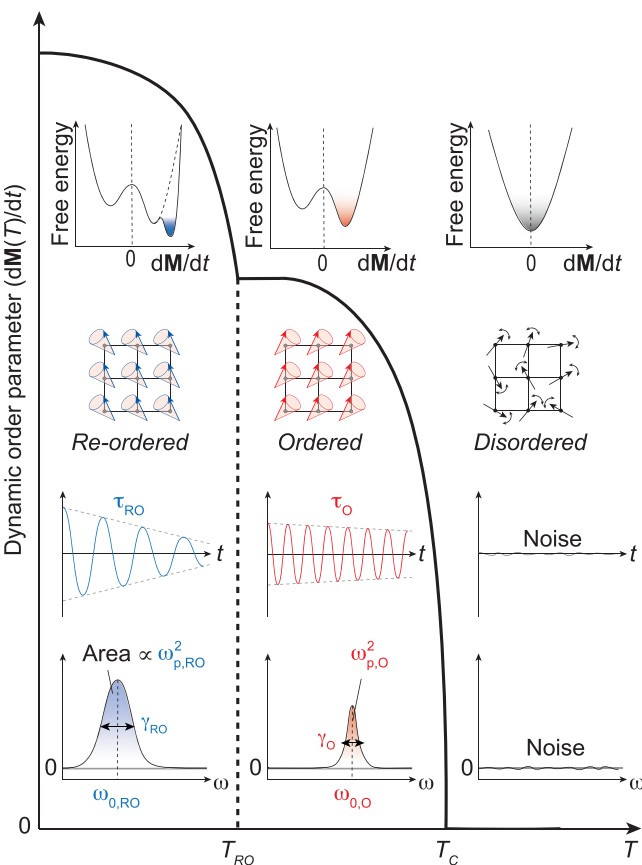

**Fig. 4 | Generalization to dynamic order parameter: serving as both order parameter and re-order parameter.** This figure illustrates the concept of a dynamic order parameter, which serves as both an order parameter and a re-order parameter Main panel: The figure demonstrates the revelation of ordering at $T_c$ (critical temperature) or re-ordering at $T_{RO}$ (re-ordering temperature) through the abrupt change in the dynamic order parameter, such as $d\mathbf{M}/dt$. First layer scheme: The corresponding Landau free energy landscape, representing the potential structure of the system. Second layer: Real-space arrangement of the order parameter, depicted as arrows on a square lattice, providing insight into the spatial distribution of the ordered state. Third layer: Time-domain signal, featuring oscillatory signals that are responsible for the dynamics of the order parameter, for example, the precession of magnetization. Fourth layer: The spectrum, showing Lorentzian peaks obtained through the Fourier transformation of the time-domain signal. The area of the spectrum is directly proportional to the dynamic order parameter, such as $\omega_p^2 \propto d\mathbf{M}/dt$.

phases in interacting thermodynamic magnets. In addition to $\Gamma_4$ to $\Gamma_1$ transition in Mn-doped YFeO$_3$, we also validate the applicability of the re-order parameter for $\Gamma_4$ to $\Gamma_2$ transition in HoFeO$_3$ (See Supplementary Fig. 6).

Our study highlights the utility of dynamic order parameters in uncovering uncharted phases and their significance in understanding physical phenomena in complex thermodynamic systems. Indeed, dynamic order parameters recently applied to phenomena such as charge density waves[24] and colloidal glasses[25].

## Methods

### Cryogenic terahertz (THz) spectroscopy
THz spectroscopic measurements were conducted on a Teraview TPS3000 (Teraview Ltd., UK) over the frequency range of 0.1–3 THz with controlling the temperature over the range of 4–300 K by a helium-free optical cryostat (Cryostation, Montana Instruments, Ltd., USA). All measurements were carried out in vacuum to remove the

water-vapor absorption. The samples were attached to a holder with a 3-mm hole by using silver paste. The THz-TDS technique yielded time-dependent waveforms of electric fields as raw data, and these were converted into complex-valued functions in the frequency domain through the fast Fourier transform (FFT).

### Sample preparation
Polycrystalline YFe$_{1-x}$Mn$_x$O$_3$ ($x = 0$–0.4) compounds were synthesized by a conventional solid state reaction method. A stoichiometric mixture of Y$_2$O$_3$ (99.98% Alfa Aesar), Fe$_2$O$_3$ (99.998% Alfa Aesar), and MnO$_2$ (99.997% Alfa Aesar) powders was ground by using a pestle in a corundum mortar, followed by pelletizing and calcining at 1100°C for 5 h. The calcined pellet was reground and sintered at 1200°C for 12 h. The compound was again finely reground and then sintered at 1300°C for 30 h and cooled to room temperature at a rate of 100°C/h. The crystallographic structure of the YFe$_{1-x}$Mn$_x$O$_3$ samples was confirmed by using an X-ray diffractometer (Ultima IV, Rigaku) with Cu-Kα radiation. The temperature and magnetic field dependences of DC magnetization were examined by a vibrating sample magnetometer for temperatures of 2–300 K and magnetic fields of –9 T to 9 T by using a physical properties measurement system (PPMS, Quantum Design, Inc.).

### Spectral analysis
We conducted the fitting of conductivity spectra using Lorentz models. (i) For lowly Mn-doped YFe$_{1-x}$Mn$_x$O$_3$ samples over $0 \le x \le 0.08$, we adopted four Lorentzian peaks, where $\omega_p$ is the oscillation strength, $\omega_0$ is the center frequency, and $\gamma$ is the broadening parameter.

$$\sigma_{total}(\omega) = \frac{\omega}{4\pi i} \frac{\omega_{p,FM}^2}{\omega_{0,FM}^2 - \omega^2 - i\gamma_{FM}\omega} \text{(FM magnon)}$$
$$+ \frac{\omega}{4\pi i} \frac{\omega_{p,AFM}^2}{\omega_{0,AFM}^2 - \omega^2 - i\gamma_{AFM}\omega} \text{(AFM magnon)}$$
$$+ \frac{\omega}{4\pi i} \frac{\omega_{p,bg}^2}{\omega_{0,bg}^2 - \omega^2 - i\gamma_{bg}\omega} \text{(Background)}$$
$$+ \frac{\omega}{4\pi i} \frac{\omega_{p,imp}^2}{\omega_{0,imp}^2 - \omega^2 - i\gamma_{imp}\omega} \text{(Impurity)}.$$

(ii) For highly Mn-doped YFe$_{1-x}$Mn$_x$O$_3$ samples over $0.1 \le x \le 0.45$, we adopted three Lorentzian peaks, where $\omega_p$ is the oscillation strength, $\omega_0$ is the center frequency, and $\gamma$ is the broadening parameter.

$$\sigma_{total}(\omega) = \frac{\omega}{4\pi i} \frac{\omega_{p,FM}^2}{\omega_{0,FM}^2 - \omega^2 - i\gamma_{FM}\omega} \text{(FM magnon)}$$
$$+ \frac{\omega}{4\pi i} \frac{\omega_{p,bg}^2}{\omega_{0,bg}^2 - \omega^2 - i\gamma_{bg}\omega} \text{(Background)}.$$

### BCS model fitting
The re-order parameter $\omega_p^2$ is fitted to the first-order-like transition with a BCS-type transition curve as below:

$$\omega_p^2(T) = A^* \tanh(((T_{RO}/T) - 1)^B) + C$$

where $T_{RO}$ is the re-ordering temperature. $A$ is a free parameter, $B$ is fixed to 0.5 or 0.8, and $C$ is a constant offset (ref. [26]).

### Data availability
The data generated in this study are provided in the Supplementary Information/Source Data file. Source data are provided with this paper.

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

## Acknowledgements

We would like to express our gratitude to J.H.K. at Yonsei University for providing access to the terahertz (THz) spectroscopy facility. This work was supported by the Samsung Science and Technology Foundation Grant (SSTF-BA2102-04), the Basic Science Research Program through the National Research Foundation of Korea (NRF-2021R1A2C3004989 and NRF-2022R1A2C1006740), and Institute for Basic Science (IBS-R011-D1). B.C.P. acknowledges the support from NRF-2019R1A6A3A01096112.

## Author contributions

T.H. and Y.J.C. guided and supervised the project. B.C.P. conceived the work. H.W.L. conducted low-temperature THz spectroscopy in the J.H.K. group. S.H.O. and Y.J.C. synthesized $YFe_{1-x}Mn_xO_3$ polycrystals and measured their magnetic susceptibilities. B.C.P. and T.H. interpreted and analyzed all the experimental data. B.C.P. and T.H. designed the data presentation. B.C.P. wrote the manuscript with input from all authors.

## Competing interests

The authors declare no competing interests.
