## [Peer Review File · Nature Communications]

REVIEWER COMMENTS

Reviewer #1 (Remarks to the Author):

As claimed by the authors, this study indeed highlights the utility of dynamic order parameters in uncovering uncharted phases and their significance in understanding physical phenomena in complex thermodynamic systems like complex spin systems (as applied before to charge density waves and colloidal glasses). The authors used Rare-earth orthoferrite YFeO₃ as a testbed for validating magnons (and their density), well known from former works about coherent control and nonlinear THz spectroscopy, as re-order parameter.

Distinct from the static equilibrium order parameter a dynamic nonequilibrium order parameter is reported, named "re-order parameter" to explain subtle phases and their transitions in the interacting ordered states. The authors used the approach to establish a phase diagram, supported by susceptibility experiments, using the re-order parameter for Mn-Doped YFeO₃ based on BCS like model fitting results. The analysis of the re-order parameter reveals spin reorientation, magnetization fluctuation, and ordering due to Mn doping, as well as magnetization compensation between phases.

The paper is scientifically and technically sound and as a generalized interesting approach, worth to be published in Nature Communications as it is because the approach to apply the re-order parameter idea to magnetic phase diagrams is new and can be applied to other materials to explain subtle changes observed in complex spin systems.

The approach in general is noteworthy for the community and of relevance for the community. The data analysis is convincing and supports the claims and conclusions at sound methodology and meets the standards of the field. There are sufficient details to reproduce the result with well known experimental techniques.

I recommend to publish the paper as it is.

Reviewer #2 (Remarks to the Author):

In the paper, the authors proposed magnon density is used as a re-order parameter to distinct the temperature dependent phase transition in Mn-doped YFeO₃. The authors also claim that a few new phases in the YFe_{1-x}MnxO₃ polycrystalline were identified with this method. I think the important contribution of the paper is the proposed “re-order parameter” (or say dynamical/nonequilibrium parameter) to identify the phase transition, that shows subtle difference from the conventional order parameter (in equilibrium state). Basically, I agree with the concept “re-order parameter” that the authors proposed, but I still have some concerns regarding the applications of the re-order parameter to identify the phase transition:

1.The phase transition between Gamma_4 and Gamma_1 with temperature for Mn-doped YFeO₃ material, the authors use BCS theory to fit the temperature dependent magnon density. I noted that the authors pick out the temperature marked with blue stars in Figure 3 a, b, and c, in which temperature the experimental data start to deviate from the BCS theoretical fitting, and the authors claim the critical temperature marks another phase transition (canted AFM to collinear AFM). Does it mean there is a new re-order parameter to describe this phase transition? If it does, what is the re-order parameter, if it does not, why?

2.I would like to mention that Y ion in YFeO₃ is not a rare-earth ion, and there is no temperature dependent spin reorientation phase transition (a second order phase transition) in YFeO₃, but for most of rare-earth orthoferrites, for example NdFeO₃, ErFeO₃, TmFeO₃ etc., the spin reorientation phase transition occurs at certain temperature range, can the spin reorientation phase transition (i.e. Gamma_2 to Gamma_4) also be identified with the proposed “re-order parameter”: magnon density? Or is this magnon density only applicable to the first order magnetic phase transition?

3.The authors use BCS theory to fit the temperature dependent magnon density, I think more explanation is needed, at least the analogue between the superconductivity phase transition and the magnetic phase transition in Mn-doped YFeO₃ should be presented.

REVIEWER COMMENTS

Reviewer #1 (Remarks to the Author):

As claimed by the authors, this study indeed highlights the utility of dynamic order parameters in uncovering uncharted phases and their significance in understanding physical phenomena in complex thermodynamic systems like complex spin systems (as applied before to charge density waves and colloidal glasses). The authors used Rare-earth orthoferrite YFeO₃ as a testbed for validating magnons (and their density), well known from former works about coherent control and nonlinear THz spectroscopy, as re-order parameter.

Distinct from the static equilibrium order parameter a dynamic nonequilibrium order parameter is reported, named "re-order parameter" to explain subtle phases and their transitions in the interacting ordered states. The authors used the approach to establish a phase diagram, supported by susceptibility experiments, using the re-order parameter for Mn-Doped YFeO₃ based on BCS like model fitting results. The analysis of the re-order parameter reveals spin reorientation, magnetization fluctuation, and ordering due to Mn doping, as well as magnetization compensation between phases. The paper is scientifically and technically sound and as a generalized interesting approach, worth to be published in Nature Communications as it is because the approach to apply the re-order parameter idea to magnetic phase diagrams is new and can be applied to other materials to explain subtle changes observed in complex spin systems.

The approach in general is noteworthy for the community and of relevance for the community. The data analysis is convincing and supports the claims and conclusions at sound methodology and meets the standards of the field. There are sufficient details to reproduce the result with well known experimental techniques.

I recommend to publish the paper as it is.

Reply) We appreciate your positive assessment. In the revision, we further aimed to advance the conceptual framework of the re-order parameter, expanding its applicability to the Γ_4 to Γ_2 transition (another form of spin reorientation) of HoFeO₃. In addition to the compelling evidence supporting the re-order parameter for the Γ_4 to Γ_1 transition in the original manuscript, the revised version systematically incorporates insights into the Γ_4 to Γ_2 transition, thereby broadening the generalization of the re-order parameter concept.

Reviewer #2 (Remarks to the Author):

In the paper, the authors proposed magnon density is used as a re-order parameter to distinct the temperature dependent phase transition in Mn-doped YFeO₃. The authors also claim that a few new phases in the YFe_{1-x}Mn_xO₃ polycrystalline were identified with this method. I think the important contribution of the paper is the proposed “re-order parameter” (or say dynamical/nonequilibrium parameter) to identify the phase transition, that shows subtle difference from the conventional order parameter (in equilibrium state). Basically, I agree with the concept “re-order parameter” that the authors proposed, but I still have some concerns regarding the applications of the re-order parameter to identify the phase transition:

Reply) We appreciate your valuable comments. In response to your concerns, we carefully prepared the revised manuscript and accompanying response letter. Kindly find our detailed point-by-point responses to your inquiries below.

1. The phase transition between Gamma₄ and Gamma₁ with temperature for Mn-doped YFeO₃ material, the authors use BCS theory to fit the temperature dependent magnon density. I noted that the authors pick out the temperature marked with blue stars in Figure 3 a, b, and c, in which temperature the experimental data start to deviate from the BCS theoretical fitting, and the authors claim the critical temperature marks another phase transition (canted AFM to collinear AFM). Does it mean there is a new re-order parameter to describe this phase transition? If it does, what is the re-order parameter, if it does not, why?

Reply) The authors appreciate for your valuable comment. We clarify that only one order parameter is necessary, yet multiple distinct transition temperatures may manifest. The individual BCS-like transitions subsequently emerge with lowering the temperature, leading to the accumulation of the re-order parameter (ω_p^2 , magnon density) as a sum. We used the re-order parameter (ω_p^2 , magnon density) for the three samples to find the phase transitions (Fig. R1). The red star in our results is the transition temperature corresponding to spin reorientation (Γ_4 to Γ_1), which is consistent with the literature and our magnetometry results. Furthermore, a blue star was identified, corresponding to a phase transition associated with the ordering of Mn. The transition temperatures (red and blue stars) were determined through BCS-type fitting to the order parameter. The established spin reorientation transition (red stars) exhibits an abrupt shift, whereas the transition linked to the blue stars appears to undergo a more gradual change.

Fig. R1 (Reviewer only) | BCS-type model fitting to the re-order parameter (ω_p^2) for Mn-doped YFeO₃ (YFe_{1-x}Mn_xO₃, x = 0.2, 0.3, 0.4). Red curves for spin reorientation transition and blue curves for subtle magnetic transition related to the Mn ordering (see main text for details).

2. I would like to mention that Y ion in YFeO3 is not a rare-earth ion, and there is no temperature dependent spin reorientation phase transition (a second order phase transition) in YFeO3, but for most of rare-earth orthoferrites, for example NdFeO3, ErFeO3, TmFeO3 etc., the spin reorientation phase transition occurs at certain temperature range, can the spin reorientation phase transition (i.e. Γ_2 to Γ_4) also be identified with the proposed “re-order parameter”: magnon density? Or is this magnon density only applicable to the first order magnetic phase transition?

Reply) Thank you for your insightful comment. Your input greatly aids in expanding the concept of the re-order parameter to spin re-orientation for rare-earth orthoferrites. We apply the re-order parameter to rare-earth orthoferrite HoFeO3 where the ionic radius of Ho is very close to that of Y.

As per your inquiry, rare-earth orthoferrites exhibit two types of spin reorientation: from Γ_4 to Γ_2 or from Γ_4 to Γ_1 . In our original manuscript, we demonstrated the application of the re-order parameter to the Γ_4 to Γ_1 transition in Mn-doped YFeO3, but did not address the Γ_4 to Γ_2 transition in rare-earth orthoferrites.

In response to your query, we grew a single crystal of rare-earth orthoferrite HoFeO3 and conducted THz spectroscopy to validate the concept of the re-order parameter for the Γ_4 to Γ_2 transition (Fig. R2). The magnon electrodynamics were measured at 2 K intervals to precisely derive the re-order parameter. The subsequent fitting to the BCS-type model provides the magnetic phase transition temperatures of ~56 K and ~38 K, consistent well with neutron scattering data [Ovsianikov, A. K. et. al. Magnetic phase diagram of HoFeO3 by neutron diffraction. *Journal of magnetism and magnetic materials* **557**, 169431 (2022)]. The phase diagram (in Fig. R2) illustrates the efficacy of the re-order parameter, generalized to the Γ_4 to Γ_2 transition in rare-earth orthoferrite HoFeO3.

In summary, the re-order parameter (we have defined) can detect not only the Γ_4 to Γ_1 transition but also the Γ_4 to Γ_2 transition. Therefore, the magnetic phase diagram can be established, together with neutron scattering experiments and the conventional magnetometry. Thanks to your insight, we had a valuable chance to generalize our re-order parameter to the Γ_4 to Γ_2 transition in rare-earth orthoferrites.

We note that new results for HoFeO3 in Fig. R2 have been inserted into Extended Data Fig. 6 in the revised manuscript with a short description as **"In addition to Γ_4 to Γ_1 transition in Mn-doped YFeO3, we also validate the applicability of re-order parameter for Γ_4 to Γ_2 transition in HoFeO3 (Extended Data Fig. 6)."** to the text on page 7. We have also decided to delete the word “rare-earth” in the revised manuscript, as yttrium is not a rare earth element.

Fig. R2 (Extended Data Fig. 6 in the revised manuscript) | Re-order parameter for HoFeO3. a, Crystal structure of HoFeO3. b, Temperature dependent re-order parameter (ω_p^2 , magnon density). Top panel for FM mode under $H_{THz} \parallel b$ -axis and bottom panel for AFM mode under $H_{THz} \parallel a$ -axis. Dots are experimental data and lines are BCS-type fitting results. Schematic illustrations are magnon modes (left figures) and the magnetic phases (top figures).

3. The authors use BCS theory to fit the temperature dependent magnon density, I think more explanation is needed, at least the analogue between the superconductivity phase transition and the magnetic phase transition in Mn-doped YFeO₃ should be presented.

Reply) We are sorry for the confusion. As you know, the BCS model can be generalized to most thermodynamic transitions, not only for superconductivity but also for magnetic phase transition. Please see the following references [Landau L.D. On the theory of phase transitions. *Journal of Experimental and Theoretical Physics* **7**, 19–32 (1937), Trenkwalder, A. et al. Quantum phase transitions with parity-symmetry breaking and hysteresis. *Nature Physics* **12**, 826–829 (2016), Liu, Y. et al., Identifying the transition order in an artificial ferroelectric van der Waals heterostructure. *Nano Letters* **22**, 1265-1269 (2022).].

The magnetic phase transition in this work is not related to the physics of superconductivity, but can be described by the phenomenologically BCS model. Note that power law in the mean-field theory (order parameter $\propto (1-T/T_c)^\beta$) is often used to describe the magnetic transitions. However, the exponent β is physically coupled to the magnetic ordering type which we do not know in this work. At least, both the power law and the BCS model find the transition temperatures in close agreement (see Fig. R3). Furthermore, the BCS model is more advantageous than mean-field theory power law: the former provides not only the transition temperature but also the saturation value reflecting the magnon density while the latter finds only the transition temperature.

However, since the term “BCS” is still confusing, we added the sentence “The magnetic transition consistently adheres to the BCS model (Methods) as well as the power law in mean-field theory (order parameter $\propto (1-T/T_c)^\beta$). However, considering the significance of the saturation value reflecting the magnon density, we opted for the BCS model. This choice not only provides the transition temperature but also furnishes the saturation value, in contrast to the power law, which only identifies the transition temperature.” on page 4-5 in the revised manuscript.

Fig. R3 (reviewer only) | Fitting of re-order parameter to BCS model (black line), compared with mean-field power law (red line) for 15% Mn-doped YFeO₃.

REVIEWERS' COMMENTS

Reviewer #2 (Remarks to the Author):

I am pleased to see the authors have addressed all my questions reasonably, and the paper has also been revised correspondingly. I would like to recommend the revised paper can be accepted as it is.